# Association of Gene Variants for Mechanical and Metabolic Muscle Quality with Cardiorespiratory and Muscular Variables Related to Performance in Skiing Athletes

**DOI:** 10.3390/genes13101798

**Published:** 2022-10-05

**Authors:** Benedikt Gasser, Martin Flück, Walter O. Frey, Paola Valdivieso, Jörg Spörri

**Affiliations:** 1Department of Sport, Physical Activity and Health, University of Basel, 4002 Basel, Switzerland; 2Laboratory for Muscle Plasticity, University of Zurich, Balgrist Campus, 8008 Zurich, Switzerland; 3Department of Medicine, University of Fribourg, 1708 Fribourg, Switzerland; 4Swiss Federal Institute of Sport—BASPO, Hauptstrasse 247, 2532 Magglingen, Switzerland; 5Hirslandenklinik, 8032 Zurich, Switzerland; 6Sports Medical Research Group, Department of Orthopaedics, Balgrist University Hospital, University of Zurich, 8008 Zurich, Switzerland; 7University Centre for Prevention and Sports Medicine, Department of Orthopaedics, Balgrist University Hospital, University of Zurich, 8008 Zurich, Switzerland

**Keywords:** ACTN3 TT, slow twitch muscle fiber, ACE, tenascin-C, PTK2

## Abstract

Background: Skiing is a popular outdoor sport posing different requirements on musculoskeletal and cardiorespiratory function to excel in competition. The extent to which genotypic features contribute to the development of performance with years of ski-specific training remains to be elucidated. We therefore tested whether prominent polymorphisms in genes for angiotensin converting enzyme (ACE-I/D, rs1799752), tenascin-C (TNC, rs2104772), actinin-3 (ACTN3, rs1815739) and PTK2 (rs7460 and rs7843014) are associated with the differentiation of cellular hallmarks of muscle metabolism and contraction in high level skiers. Material & Methods: Forty-three skiers of a world-leading national ski team performed exhaustive cardiopulmonary exercise testing as well as isokinetic strength testing for single contractions, whereby 230 cardiopulmonary measurements were performed in the period from 2015–2018. A total of 168 and 62 data measurements were from the Alpine and Nordic skiing squads, respectively. Ninety-five and one hundred thirty-five measurements, respectively, were from male and female athletes. The average (±SD) age was 21.5 ± 3.0 years, height 174.0 ± 8.7 cm, and weight 71.0 ± 10.9 kg for the analysed skiers. Furthermore, all skiers were analysed concerning their genotype ACE-I/D, Tenascin C, ACTN3, PTK2. Results: The genotype distribution deviated from Hardy–Weinberg equilibrium for the ACTN3 genotype, where rs1815739-TT genotypes (corresponding to the nonsense mutation) were overrepresented in world-class skiers, indicating a slow muscle fibre phenotype. Furthermore, the heterozygous rs2104772-AT genotypes of TNC also demonstrated the best scaled peak power output values during ramp exercise to exhaustion. The highest values under maximum performance for heart rate were associated with the rs1799752-II and rs1815739-CC genotypes. The lowest values for peak power of single contractions were achieved for rs1815739-CC, rs1799752-II and rs7843014-CT genotypes. The skiing discipline demonstrated a main influence on cardiorespiratory parameters but did not further interact with genotype-associated variability in performance. Discussion: Classically, it is pointed out that muscles of, for example, alpine skiers do not possess a distinct fibre type composition, but that skiers tend to have a preponderance of slow-twitch fibres. Consequently, our findings of an overrepresentation of ACTN3-TT genotypes in a highly selective sample of elite world class skiers support the potential superiority of a slow fibre type distribution. Conclusions: We suggest that one competitive advantage that results from a slow, typically fatigue-resistant fibre type distribution might be that performance during intense training days is better preserved, whereby simply a higher technical training volume can be performed, yielding to a competitive advantage.

## 1. Introduction

Alpine and Nordic skiing places various demands on the musculoskeletal and cardiorespiratory systems [1,2,3]. Thereby, performance capacity is defined by the anatomic design of functional structures depending on genetic influences, whereby high demands exist for the cardiovascular and musculoskeletal systems [4,5,6,7,8,9,10,11,12,13]. In addition to an important role of aerobic energy combustion, especially in the downhill phases of skiing, strength and proprioceptive skills play an important role [9,10,11,12,13]. Despite the carving skis in the 1990s, alpine skiing and, in particular, its competition discipline giant slalom, is still defined by a considerable amount of flexion-extension movement [7,13,14,15]; especially when passing the gate, a highly angulated body position is necessary [13]. The resulting forces must be dosed to maintain balance and to find the fastest line to the next gate. The same applies in downhill parts of cross-country skiing, in which a high and intermittent mechanical load on the leg muscles during turning actions is released [1,2,9,10,11,12,13]. Therefore, the encountered movements, which are typically slow compared to other types of sports, comprise a considerable degree of eccentric loading for longer durations to preserve and stabilize the centre of mass [1,2,9,10,11,12,13]. The force pattern used is dependent on a high share of eccentric muscle work (highest in technical disciplines of alpine skiing), implying that the muscle is stretched and absorbs mechanical energy that is either transformed into heat or is stored like a feather as elastic energy [9]. 

A significant relationship between the dosage of eccentric muscle action and ski-specific performance was identified [9]. Consequently, alpine skiers are typically characterized by rather strong leg extensor and flexor muscle groups that are primarily slow [1,2,9,10,11,13,16]. In Nordic skiing, with its different disciplines cross-country skiing, telemark, ski-jumping or a combination of cross-country skiing and ski-jumping are sometimes similar, and yet sometimes different cardiovascular and muscular requirements occur in comparison to Alpine. As cross-country skiing is considered one of the most demanding sports concerning aerobic capacity, specific adaptations of the cardiac and musculoskeletal systems can be identified [17]. Despite the fact that there is a considerable amount of differences in Alpine versus Nordic skiing, similar requirements for stabilizing and controlling acceleration of the body when gliding and turning can be identified [1,2,9,10,11,13]. This coordinative ability is dependent on a good cardiovascular and muscular fitness of the athletes [1,2,9,10,11,13]. Thereby, in both Alpine and Nordic skiers, the physiological characteristics of the elite skier are a result of both genetics and adaptation to rigorous multiyear, year-round training programmes [17]. Consequently, well-trained athletes demonstrate the specialization of metabolism- and contraction-related cellular parameters compared to normal subjects depending on gene polymorphism [4]. Differences in cellular parameters could be identified for genotypes ACE-I/D (insertion/deletion), rs1799752) and Tenascin TNC (A/T, rs2104772), and Actinin ACTN3 (C/T, rs1815739) and localized post hoc when taking the interaction with the phenotype into account [4]. Between the endurance and power of well-trained athletes, these concerned effects on capillary length density for the aforementioned ACE-I/D and TNC genotypes, fibre type distribution and volume densities of myofibrils for ACE-I/D genotypes and fibre mean a cross-sectional area (MSCA) for the TNC genotype [4]. Endurance athletes, carrying the I-allele of ACE (I/D), demonstrated 50%-higher volume densities of mitochondria and sarcoplasma, whereas power athletes that carried only the D-allele showed the highest fibre MCSAs and a lower percentage of slow-type muscle fibres [4]. The differentiation of phenotypic traits with systemic training, especially regarding cardiovascular and muscular features, is understood to be affected by genetic factors [4,18].

Natural variations in gene sequences are known to be associated with differences in structural and functional aspects of muscle-based performance [4,18]. For instance, polymorphisms in the genes for ACTN3 (rs1815739) and PTK2 (protein tyrosine kinase 2; rs7460 and rs7843014) are associated with higher values and gains in muscle strength with resistance type training [4,18]. On the other hand, ACE and TNC-related genotypes, rs1799752 and rs2104772, are associated with different improvements in aerobic metabolic features and are overrepresented in power athletes vs. endurance athletes [19,20]. Plausibly, preferred combinations of gene variants that mainly affect mechanical (actinin-3, PTK2) and metabolic (ACE, tenascin-C) aspects of skeletal muscle and cardiovascular functioning are likely to affect skill levels in elite skiers [4,19,20,21,22,23]. Therefore, the proteins encoded by the actinin-3 (ACTN3) and PTK2 genes localize to protein complexes that stabilize muscle fibres and “translate” mechanical forces into biochemical signalling processes and should show a specific pattern in athletes [24,25]. Conversely, the proteins encoded by the ACE and tenascin-C (TNC) genes operate in the same pathway [24,25]. Thus, it is expected that ACE × TNC genotypes demonstrate (in part) synergistic effects [14,22,23] on skeletal muscle and cardiovascular functioning that are likely to affect the skill level in elite skiers [4,19,20,21,22,23]. Currently, however, little is known about the possible effect sizes of the aforementioned genetic factors on performance factors in elite skiers.

Accordingly, in the current study, we investigated the extent to which peak values for muscle strength and indices of cardiovascular performance are associated with sequence variants in the genes for ACTN3, PTK2, ACE-I/D and TNC. The strength of possible genetic associations was assessed based on effect sizes. As a hypothesis with potential falsification, we stated that there is no effect of the genotype on aspects of skeletal muscle and cardiovascular function [26].

## 2. Material & Methods

### 2.1. Subjects

Forty-three alpine and Nordic skiers (here, solely biathletes or cross-country skiers) of a national team of a world class skiing country participated in the study. Most subjects typically originate from rural areas in higher areas of Switzerland. Cardiopulmonary exercise (CPX) testing in pre- and postseason and peak performance of single muscle contractions was performed. For alpine skiers, CPX was performed on a cycling ergometer, whereas for Nordic skiers, CPX was performed on a treadmill. In total, 230 data points were available, whereby a total of 168 and 62 data points were from alpine and Nordic skiers, respectively. Ninety-five and one hundred thirty-five measurements, respectively, were from male (21 subjects) and female (22 subjects) athletes. The average age was 21.5 ± 3.02 years (Table 1).

### 2.2. Ethics

The study involving human participants was reviewed and approved by the Ethics Committee of the Canton of Zurich (Switzerland)-BASEC Number: 2018-01598. The investigation (Project title “Untersuchung zum Zusammenhang genetischen Faktoren und dem muskelabhängigen Phänotyps”- GEMUP) was conducted according to the principles expressed in the Declaration of Helsinki. Informed consent, written and oral, was obtained personally from each of the study participants.

### 2.3. Design

All subjects performed exhaustive CPX testing and isokinetic strength testing over 4 years from 2015–2018, yielding 230 data points of CPX in female and male, alpine and Nordic skiers. Genotypes for five selected gene polymorphisms were determined posthoc and assessed for associations with the measured physiological parameters and the reported genotype distributions [4].

### 2.4. Peak Performance of Single Contractions

The peak performance of single contractions was measured with a CON-TREX^®^ isokinetic dynamometer (multijoint module 2100 CON-TREX-MJ with power module 1100 CON-TREX-PM-1, CMV AG, Dübendorf, Switzerland). The peak performance of single contractions was measured for flexion and extension strength for isokinetic single contractions at 360° s^−1^. The device was calibrated per the manufacturer’s specifications and was verified prior to testing, and the measurement procedure was performed according to a certified protocol by the Swiss Olympics. Strength tests were typically performed after a warm-up on the same day prior to the CPX testing.

### 2.5. Aerobic Performance

CPX was performed as a ramp exercise on a cycling ergometer (alpine skiers) or treadmill (Nordic skiers). The test protocol was conducted in an air-conditioned laboratory at a standardized temperature of 20 °C according to a modified version of a published protocol by Swiss Olympic [29]. In brief, the test started with at least 2 min of rest, without a specific warm-up, when subjects sat still in an upright sitting position in an air-conditioned laboratory on an electrically braked cycle ergometer (Ergoselect 200, Ergoline, Bitz, Germany). Subjects were asked to maintain a normal breathing pattern. In addition, blood lactate concentration was measured by collecting a sample of blood from the earlobe every 2 min using a lancing device (Akku-Check, Safe-T-Pro-Plus, Roche Diabetes Care) and a blood lactate monitor (Biosen C-Line, EKF-diagnostic). During the entire duration of the exercise test, subjects were advised and supervised by an experienced sports scientist. The test was stopped when the subjects experienced volitional exhaustion and/or were not able to maintain the target pedal cadence and power output. Subsequently, recordings continued for a period of eight min, when subjects rested in a seated position on the cycle ergometer. In Nordic skiers, CPX testing was performed on a treadmill Woodway PPS Sport (Woodway GmbH, Weil am Rhein, Germany). The start was either at 5.4 km/h, 7.2 km/h or 8 km/h (depending on training state) with an increase of 1.8 km/h every 3 min. The start was with an angle of 2 percent of the treadmill, which was not altered until 14.4 km/h and then continuously increased. Cardiorespiratory parameters were continuously monitored with a Metalyzer 3B-R2 device (Cortex, Leipzig, Germany), which was accompanied by the assessment of systolic and diastolic blood pressure (Tango+, SunTech Medical Inc., Morrisville, NC, USA), heart rate (Suunto t6d, Suunto, Vantaa, Finland), SpO_2_ (arterial oxygen saturation), heart rate and minute ventilation (VE) in [L min^−1^]. The rate of perceived exertion was assessed each second minute with the Borg scale [30].

### 2.6. Genotyping

Genotyping was performed based on the specific amplification of genomic DNA sequences for gene polymorphisms of interest (ACE-I/D, rs1799752; ACTN3, rs1815739; TNC, rs2104772; PTK2, rs7460 and rs7843014) with polymerase chain reactions (PCR), and the subsequent detection of genetic variation was performed by melting curve analysis. Where indicated, the genotype was confirmed by sequencing and gel electrophoresis. 

In detail, blood was drawn from the cubital vene, and genomic DNA was extracted by the DNeasy Blood and Tissue Kit (Cat. No 69504, Qiagen, Basel, Switzerland) according to the manufacturer’s instructions. DNA concentration and purity were measured using a NanoDrop USV-99 AGTGene (Labgene Scientific, Châtel-St-Denis, Switzerland). Absorbance measurements at 260 nm and 280 nm indicated that the final DNA concentration ranged from 10 to 50 ng/μL and that the DNA was of high purity. DNA samples were diluted to a final concentration of 5 ng/μL and stored at −20 °C until analysis. Subsequently, PCRs of multiple samples were conducted in duplicate in a volume of 10 μL with 10 ng genomic DNA in 48-well plates with a specific combination of primers (0.2 μM) and 2.5 mmol MgCl2 in 1x KAPA HRM FAST Master Mix [KAPA BIOSYSTEMS, Labgene Scientific, Châtel-St-Denis, Switzerland]. PCR-based amplification reactions and subsequent high-resolution melting (HRM) curve analysis were run with a real-time PCR system (EcoTM, Illumina, San Diego, United States, distributed by Labgene Scientific, Châtel-St-Denis, Switzerland). Subsequently, a genotype analysis was carried out using genetic variation analysis software (EcoStudy Version 5.0, Illumina R, San Diego, USA). The generic setting was as follows: 3 min enzyme activation at 95 °C, followed by 35 amplification cycles (5 s denaturation at 95 °C and 30 s annealing/extension at 60 °C), and a final melting cycle (initiated by heating to 95 °C and a thermal ramp after cooling to 55 °C to 95 °C).

For the amplification and detection of the ACE-I/D gene polymorphism rs1799752, two different primer pairs were used as described [18]. For the 66-bp-long amplicon of the I-allele primers ACE2 (5′-tgggattacaggcgtgatacag-3′) and ACE3 (5′ atttcagagctggaataaaatt-3′) were deployed. For the 83-bp-long amplicon, the D-allele primers ACE1 (5′-catcctttctcccatttctc-3′) and ACE3 (5′-atttcagagctggaataaaatt-3′) were used. The amplification and HRM-based detection of sequence variation in an 85-bp-long DNA fragment for rs2104772 was conducted with primers 5-caaaaagcagtctgagccac-3′ and 5-ttcagtagcctctctgagac-3′) as described [31]. Equally, the amplification and detection of a 291-bp-long DNA fragment for rs1815739 with HRM-based PCR was carried out as described [4] with forward (5′-ctgtttgcctgtgtgtaagtggggggg-3) and reverse (5′- tgtcacagtatgcaggagggg-3′) primers. Similarly, pairs of forward and reverse primers were used for amplifying and detecting rs7460 (5′-tgggtcgggaactagctgta-3′, 5’-atggaaaaaggggatggtcc-3’) and rs7843014 (5’-tgatgggacctaaacccatt-3’, 5’-tttcccatcagctgcttgtt-3’) specific DNA fragments.

The specificity of the PCRs was validated in experiments with negative and positive controls and by confirming the predicted PCR product size via agarose gel electrophoresis. This was followed by DNA band isolation and subsequent DNA sequencing by Microsynth (Balgach, Switzerland). The negative control was a nontemplate control reaction (NTC) with DNAse-free water, and positive controls comprised genotyped DNA samples, including samples for each genotypic variant, which were confirmed by microsequencing. The positive controls also served as internal references to generate a melting profile for screening unknown samples. For each experimental run, raw data of the melting curves were normalized versus an internal reference for a given genotype to reveal the genotype of a given sample.

### 2.7. Data Handling

For the physiological parameters, such as VO_2_ peak or heart rate, in addition to raw values, scaling was adapted. Values were scaled to body mass to account for differences in body size and mass and normalise the data distribution. Values derived from CPX testing were divided by body mass. Values derived from dynamometer measurements were scaled exponentially to body mass as described [32]. with different exponents for peak power and moment during extension (i.e., 0.70 and 1.39) and flexion (0.56, 1.01) [33]. Calculation of slopes vs power output for respiratory parameters during the ramp test was conducted as described in [34].

### 2.8. Statistics

A two-way ANOVA (2 × 2) for factors genotype (maximally two factors were assessed at once) × sex × discipline, and optionally the discipline (alpine, nordic). Hardy-Weinberg equilibrium was calculated using a downloadable tool based on a Chi^2^-Test [35]. Repeated measures ANOVA for the repeated factors contraction modality (eccentric, concentric), velocities (360° sec^−1^, 60° sec^−1^) and leg (left, right) and the factors genotype (maximally two assessed at once) × sex × discipline was conducted to test effects on peak power (and moment) of single contractions. For the ramp exercise to exhaustion, univariate analyses of variance (ANOVAs) with post-hoc tests for the least significant difference were run to identify the effects of the factors “aerobic fitness” × “ACE-I/D” genotype on the r-values and slopes of linear correlations during ramp exercise. Genotype-specific differences were calculated based on a dominant genetic model for the ACE D-allele (I/I vs. I/D vs. D/D). Differences with *p* < 0.05 were considered significant. The Shapiro-Wilks test, with its strong power for small samples, was used to analyse the normal distribution in the measured and derived parameters. In cases where a normal distribution could not be assumed, a Welch ANOVA was conducted to confirm genotype effects being detected based on a univariate ANOVA. Finally, values were displayed in box-whisker plots or histograms in MS-Excel (Microsoft Office 365). Effect sizes were assembled using MS-Excel (Microsoft Office 365) and exported into Treeview (http://jtreeview.sourceforge.net, accessed on 17 April 2022) to be displayed in colour code (Appendix A).

## 3. Results

Figure 1 and Table 1 and Table 2 show the physiological characteristics and the genotype distribution of the participating elite skiers and reference populations. In contrast to the normal population or strength athletes, the distribution of the genotypes for the studied rs1815739 polymorphism in the ACTN3 gene deviated from Hardy-Weinberg equilibrium for the skiers. Thereby, the rs1815739-TT genotype was overrepresented in skiing athletes with respect to the rs1815739 genotypes holding the C-allele (Figure 1).

Figure 2 and Appendix A depict the observed effect sizes and *p*-values. The strongest associations with an effect size above 0.1 were identified for sex and skiing discipline and certain gene polymorphisms. Regarding the latter, the gene polymorphisms rs1815739, rs1799752, and rs7843014 were associated with variability in peak power (contraction modalities and velocities combined), whereas the gene polymorphism rs1815739 was associated with differences in peak torque (contraction modalities and velocities combined). Concerning the performance during CPX-testing, the strongest associations were observed for peak power output and gene polymorphism rs2104772, and for peak heart rate and the interaction between rs1815739 × rs1799752 gene polymorphisms.

Concerning the strength parameters, Figure 3 shows box-whisker plots of the peak performance during isokinetic contractions for genotypes rs1815739, rs1799752 and rs7843014. Thereby, posthoc differences for peak power output were identified between carriers and noncarriers of the T-allele of the ACTN3 gene polymorphism rs1815739 in extension (CC vs. CT, *p* = 4.6 × 10^−19^ CC vs. TT, *p* = 4.0 × 10^−28^), and flexion (CC vs. CT, *p* = 7.7 × 10^−13^; CC vs. TT, *p* = 5.6 × 10^−19^), and homozygous and heterozygous genotypes of ACE-I/D (rs1799752) in extension (II vs. DD, *p* = 4.0 × 10^−5^; II vs. ID, *p* = 1.6 × 10^−7^), flexion (DD vs. ID, *p* = 9.2 × 10^−4^; DD vs II, *p* = 4.0 × 10^−6^) and PTK2 gene polymorphisms (rs7843014) in extension (CC vs. CT, *p* = 1.7 × 10^−4^; TT vs. CT, *p* = 5.7 × 10^−4^) and flexion (CC vs. CT, *p* = 0.029; TT vs. CT, *p* = 0.006), respectively.

Figure 4 shows the box-whisker plots of the identified main associations and post-hoc effects of parameters of aerobic capacity with individual genotypes. For the TNC gene polymorphism rs2104772, the heterozygous AT genotypes demonstrated the best values for body-mass scaled VO_2_ peak (AT vs. AA, *p* = 3.0 × 10^−6^; AT vs. TT, *p* = 8.5 × 10^−12^), whereas the peak respiration rate (RR) was highest in homozygous A-allele carriers (AA vs. AT, *p* = 5.0 × 10^−4^; AA vs. TT, *p* = 3.0 × 10^−9^). For PTK2 gene polymorphism rs7843014, the heterozygous CT genotype demonstrated the lowest values for peak VE (CT vs. CC, *p* = 3.6 × 10^−6^; CT vs. TT, 1.1 × 10^−12^) and peak VT during CPX-testing (CT vs. CC, *p* = 0.813; CT vs. TT, 6.2 × 10^−9^).

Figure 5 highlights the association of differences in peak performance with the interdependence of the two genotypes. The highest values for maximum heart rate were achieved for rs1815739-CC genotypes with a homozygous genotype for the ACE I-allele (rs1799752; II vs. ID, *p* = 7.5 × 10^−16^; II vs. DD, *p* = 2.0 × 10^−13^). Conversely, the scaled peak power output was lowest in the homozygous carriers of the ACE I-allele in rs1815739-CC genotypes (II vs. DD, *p* = 6.0 × 10^−6^; II vs. ID, *p* = 6.0 × 10^−6^).

## 4. Discussion

The aim of this study was to elucidate the effects of genotype on performance characteristics in elite skiing athletes from an elite Alpine and Nordic skiing squad. The initially stated hypothesis that there is no association between gene polymorphism and functional parameters of skeletal muscle and the cardiovascular system can be rejected [26]. Here, we identified an effect of the assessed ACE and Tenascin polymorphism on cardiovascular parameters. At the skeletal muscle level, strong effects of the studied genotypes on ACTN3, PTK2 and ACE was detected based on effect sizes > 0.1, which contrasts with previous findings [24,25,36,37]. Our observations emphasize that discrete and considerable associations exist for distinct elements of genetic variation and parameters of the performance phenotype. Thereby, it has to be kept in mind that we assessed a highly selective sample, where some of the subjects analysed are one of the best skiers worldwide and all were members of the elite skiing squad.

One key finding is that the genotype distribution deviated from Hardy-Weinberg equilibrium for the ACTN3 genotype rs1815739, where TT genotypes (corresponding to the nonsense mutation) were overrepresented (Figure 1). Interestingly, in 1999, the common single-base transversion (C¡T) in exon 16 of the ACTN3 gene that converts an arginine residue (R) to a stop codon (X) at amino acid position 577 was described [38,39]. Approximately 16% of the world’s population is completely deficient in α-actinin-3 protein due to homozygosity for the R577X stop codon (ACTN3 577XX genotype) [38,40]. There is variation in the frequency of the R577X allele in different ethnic groups, with allele frequencies of 0.55 in Europeans, 0.52 in Asian populations, and 0.09 in Africans [38,41]. To point out, most of the skiers participating in our study came from higher areas in the Alps, which might also explain the detected distribution in our sample. Interestingly, hints from a mouse model indicate that grip strength is significantly lower in ACTN3 KO mice (6–7%) than in WT mice [40]. Intriguingly, it was found that ACTN3 KO mice have an increased capacity to run longer distances [41]. These data are consistent with the findings of an original human association study in which a trend toward an increase in the frequency of XX individuals among endurance athletes was found, reaching significance among female athletes [38]. This association was also significant when road cycling athletes were analysed separately and in a study of Israeli elite athletes [38,42,43]. However, other studies have not replicated the association between the ACTN3 genotype and elite endurance performance [38]. This suggests that the association between α-actinin-3 deficiency and endurance is not as strong as its association with reduced performance in sprint and power activities [38,44,45,46,47,48,49].

When focusing back on our results, the finding of a slow muscle phenotype stands in contrast to those from Estonia, whereby skiers showed higher frequencies of the rs1815739-CC genotypes, indicating a fast fibre distribution in young male skiers compared to healthy controls [50]. Nevertheless, classic evidence indicates that muscle fibre distribution of alpine skiers does not possess a distinct fibre type composition and, if anything, skiers tend to show a preponderance of slow twitch fibres in line with our findings of a dominance of ACTN3-TT genotypes [1]. Skiing comprises many postural and slow-type contraction movements, which in principle supports the findings of a predominance of a slow fibre type distribution [1]. Heritable factors arguably play a role in athletic performance by setting upper limits and directions for tissue adaptations with training, which intuitively has relevance for elite skiers. An attempt to understand the distribution of certain genotypes in this group of sportsmen has been conducted with participants in Estonia [50], Poland [51] and Turkey [52]. In a study from Estonia, the frequencies of the rs1799752 and the rs1815739-CC genotypes stand in contrast to our investigation and the frequency of the rs1815739-CT genotype was lower in young male skiers than in controls [50]. A study from Poland implied that ACE gene polymorphism-related variation was not a determinant of aerobic capacity in either female or male, well-trained endurance athletes participating in winter sports [51]. A study from Turkey investigated a single nucleotide polymorphism in the gene for hypoxia-inducible factor 1 α [52], which is an activity-dependent transcriptional regulator of the slow oxidative muscle phenotype that we did not assess, making direct comparisons impossible. Nevertheless, HIF-1a is an activity-dependent transcriptional regulator of the slow oxidative muscle phenotype (reviewed in Däpp et al. [53]), whereby C-allele carriers of rs11549465 affect the stability of the HIF1A protein and were shown to be enriched in endurance type athletes [54]. To conclude, the analyses from Turkey implied that a HIF-1a related genetic influence exists which is expected to favour the slow oxidative muscle phenotype, in line with our findings of an overrepresentation of rs1815739-TT genotypes [52].

Furthermore, post-hoc differences underpinning the large effect size for the association between the studied ACTN3 single nucleotide polymorphism and peak performance are in line with a favourable influence of the slow muscle fibre phenotype on power development during single contractions [55]. However, the higher peak power values in T-allele carriers contrast with the common expectation, where homozygous C-allele carriers are expected to demonstrate a higher increase in peak torque than homozygous T-allele carriers [56]. Skiers often show a large degree of slow-type muscle fibres [1], which may demonstrate hypertrophy with slow isometric-type ski training (reviewed in Alhammoud et al., 2020) [7]. Our findings therefore possibly reflect the discipline-specific hypertrophy of slow-type fibres.

The unusual enrichment of the rs1815739-TT genotype for ACTN3 in the investigated skiers, some of which belong to the current best worldwide, still poses some questions. In principle, this implies a need for a slow muscle fibre composition. Looking for reasons, one can mention the need for hard training in the summer of elite athletes exposing the body to a considerable amount of aerobic work to absorb the hard discipline-specific training in autumn and winter. Thereby, specific genetically primed training effects occur. For example, the sectional area (CSA) of m. vastus lateralis and the mean CSA of slow-type fibres, which correlated with peak power output after endurance training interacted with the ACTN3 genotype [4]. Intriguingly, the rs1815739-TT genotype is also related to a better tolerance of prolonged altitude exposure which the studied elite athletes typically encounter(ed) during their adolescence and training [57].

Genotype × training interactions in muscle also resolved for ACE transcript levels and by a trend for the pro-angiogenic factor tenascin-C post exercise [18]. The observations indicated that variability in aerobic performance (in the studied subjects) was in part reflected by an ACE-I/D-genotype-modulated metabolic phenotype of a major locomotor muscle. Furthermore, repeated endurance exercise appeared to override the genetic influence in skeletal muscle by altering the ACE-related metabolic response and molecular aspects of the angiogenic response to endurance exercise [18]. To be sure, in our sample we clearly detected associations between ACE polymorphism and performance parameters in our investigation in contrast to the findings from Poland [51] and Estonia [50] (for details, see above). From a metabolic point of view, a day of intense skiing training on an icy hill requires a high rate of glycogen utilization that eventually may result in depletion of muscle glycogen stores by the end of the day [1]. We suggest that from a metabolic point of view, slow muscle fibres are simply better suited for skiers training on a glacier when preparing the next season, as simply technical training can be performed for a longer duration [1]. When analysing direct effects for single contractions (Figure 3) lowest values were achieved for rs1815739-CC and rs7843014-CT genotypes in extension and rs1799752-II genotypes in flexion. This implies for the rs1815739-CC genotype that a fast muscle fibre composition shows a lower maximum power output. These findings are supported by current evidence showing that skiing places a high and rather frequent mechanical demand on leg muscles during turning actions [1,2,9,10,11,12].

Focusing on aerobic capacity for individual genotypes, post-hoc effects were identified between heterozygous and homozygous A- or T-allele carriers of rs2104772 for body-mass scaled (i.e., related) peak VO_2_, peak respiration rate and peak power during the ramp test to exhaustion. This indicates that this genotype for TNC must have a small but robust effect on performance parameters [21]. Corresponding to high values of scaled peak VO_2_, the heterozygous AT carriers of rs2104772 also demonstrated the best values for the scaled peak power output during the ramp exercise to exhaustion. The identified order of differences was reminiscent of the influence of the studied TNC genotype on capillary length density in extensor muscle [31]. This might be interpreted to mean that TNC protein influences capillary length density in extensor muscles, possibly reflecting a trade between the promotion of angiogenesis post exercise in T-allele carriers of rs2104772 with lower TNC expression, and an expected lower lung volume [21]. Furthermore, Figure 4 shows the parameters of performance for combined genotypes of rs1815739 and rs1799752. Maximal heart rate during the ramp test to exhaustion is dependent on the genetic influence being highest for the homogenous I-allele carriers in the ACE gene (i.e., rs1815739-II) in rs1815739-CC genotypes. Collectively, the findings corroborate the view that a genetic endowment of modern skiers with genetic features that are associated with favourable adjustments of the slow oxidative muscle phenotype. This may have its basis in the rather slow type of contractions that are a characteristic of most skiing disciplines on snow [1,2,9,10,11,12] and the fact that the muscles of these athletes require a rather high level of activity to stabilize posture.

The further observed association between the studied genotypes for PTK2 (rs7843014) and ACE (rs1799752) and muscle power revealed a major influence on the exercise phenotype. This finding possibly reflects the hypertrophy of muscle fibres which is demonstrated to prevail for the rs7843014-TT and rs1815739-TT genotypes (i.e., Figure 3). Intriguingly, we also observed a strong association between the rs2104772-genotype and body-related VO_2_peak, which was explained by the largest values in the ‘intermediate’ AT genotype. This observation is related to similar trends for capillary density in the knee extensor muscle of power athletes [4]. Intriguingly, lung volumes may also be affected by TNC [21]. As the studied SNP [31] affects TNC levels, our findings arguably highlight a regulatory role of TNC in the growth of endothelial structures that line the capillary three and inner surface of the lung.

A number of limitations may be considered when evaluating our results. For instance, we scaled raw values to body mass to account for differences in body size and normalise the data distribution described [32]. Nevertheless, sex and the skiing discipline was noted as the most influential factor (Figure 2), but it did not essentially dissolve the main effects (Figure 2 and Appendix A). Additionally, we note that certain measures stem from biological replicates over progressive years of training to reinforce the reproducibility of possible effects but without that we applied a statistical correction. Last, we refer to the aspect that only main effects were considered for the display of post-hoc differences. We therefore missed showing relevant differences in graphs such as those observed for body-mass-related peakVO_2_ between rs1799752-II and rs1799752-DD genotypes of ACE (4 mL O_2_ min^−1^ kg^−1^, *p* = 0.003).

## 5. Conclusions

To summarize, as classic evidence pinpointed that muscles of alpine skiers do not possess a distinct fibre type composition, but that skiers tend to have a preponderance of slow-twitch fibres, our findings of an overrepresentation of the rs1815739-TT genotype for the nonsense mutation of ACTN3 support the potential superiority of a slow fibre type distribution in high level alpine skiers. The extent to which this relates to the reportedly better coping with effects of prolonged altitude exposure for rs1815739-TT genotypes, which is typically encountered for most for the studied individuals during their adolescence and as part of their training regimes, remains to be explored [57]. We suggest that one competitive advantage that results from a slow fibre type distribution might be during intense training days, whereby simply a higher technical training volume can be performed. However, as muscle tissue plasticity is controlled via complex, interdependent and partially redundant signal networks, which have multiple entry points [58], we must assume that molecular networks of interactions, in addition to those being addressed with our genetic approach, must be considered to reveal an encompassing understanding of variability in phenotypic reactions to years of training in skiers. The herein studied associations between genotype × performance phenotype might help to resolve the enigma of these complex processes.

## Figures and Tables

**Figure 1 genes-13-01798-f001:**
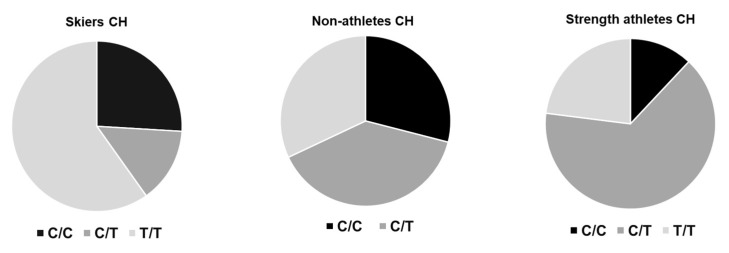
Genotype distribution across alpine and cross-country skiing. Pie charts depicting the frequency of the three genotypes of the ACTN3 gene polymorphism rs1815739 for the studied elite skiers and controls from previous studies with strength athletes and nonathletes from the same country [27,28].

**Figure 2 genes-13-01798-f002:**
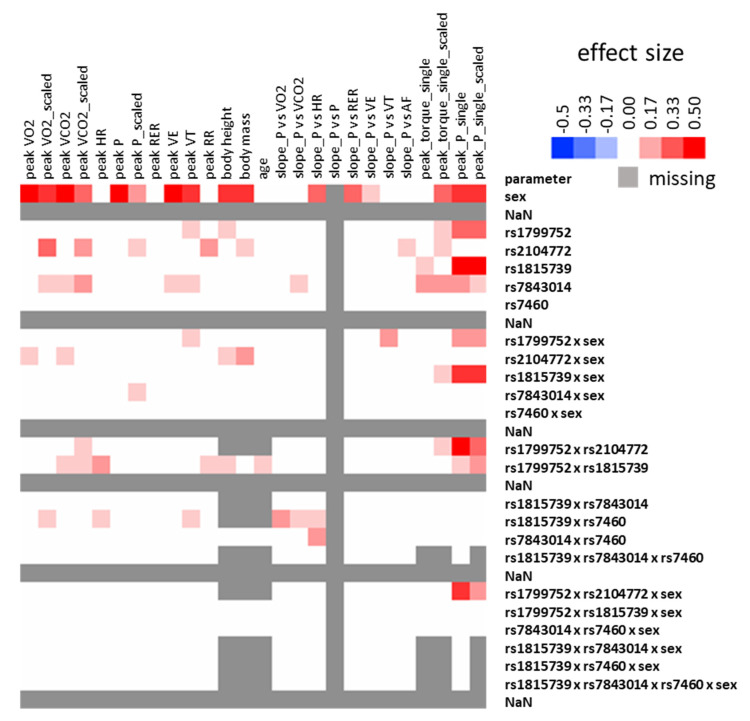
Heatmap of effect sizes. Coloured graph visualizing the calculated effect sizes for cardiorespiratory parameters during the CPX testing and performance values during the dynamometer test as calculated with ANOVAs for the factors genotype × sex × discipline. Effect sizes above 0.10 are coloured in red, with those below appearing in white, or if calculation was not possible in grey. Abbreviations: HR, heart rate; NaN, no values; peak P, peak power during CPX testing; peak_P_single, peak power of a single contraction; peak_torque, peak torque of a single contraction; RR, respiration rate; rs1799752, genotype rs1799752 (ACE-I/D); rs1815739, genotype rs1815739 (ACTN3); rs7640, genotype rs7640 (PTK2); 7843014, genotype rs7843014 (PTK2); rs2104772, genotype rs2104772 (TNC); RER, respiration exchange ratio; slope, slope of the relationship vs power output during CPX testing; scaled, body-mass related values; VCO_2_, systemic carbon dioxide production; VE, minute ventilation; VT, tidal volume; VO_2_, systemic oxygen utilization; x, indicates interactions between listed factors.

**Figure 3 genes-13-01798-f003:**
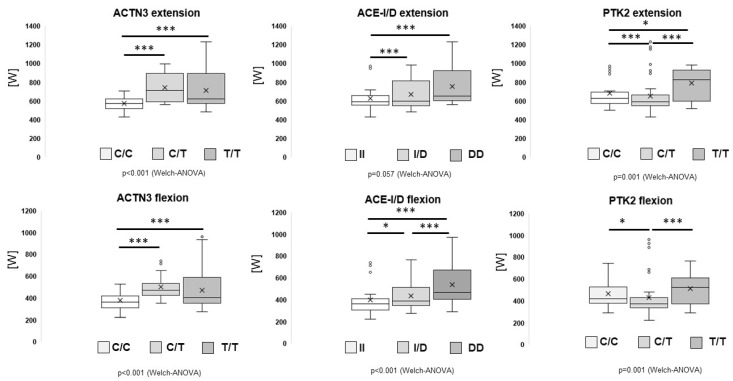
Main genotype associations with strength. Box-whisker plots for peak power during isokinetic single contractions at 360° sec^−1^ in extension and flexion for the studied genotypes for the ACTN3 (rs1815739), ACE-I/D (rs1799752) and PTK2 (rs7843014) gene polymorphisms. Repeated measures ANOVA with post hoc test of least significant difference. The corresponding p-values for the Welch type ANOVA are indicated below each panel. * and *** indicate, *p* ≤ 0.05 and 0.001, respectively, for the indicated comparisons.

**Figure 4 genes-13-01798-f004:**
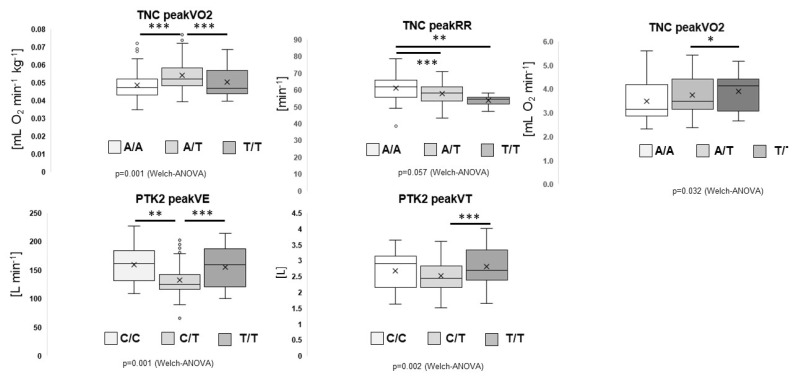
Main genotype associations with cardiorespiratory parameters. Box-whisker plots for absolute and body-mass-related peak VO_2_ and peakRR for TNC (rs2104772) genotypes and peakVE and peakVT for PTK2 (rs7843014) genotypes. ANOVA with post hoc test of least significant difference. The corresponding *p*-values for the Welch type ANOVA are indicated below each panel. *, ** and *** indicate, *p* ≤ 0.05, 0.01 and 0.001, respectively, for the indicated comparisons.

**Figure 5 genes-13-01798-f005:**
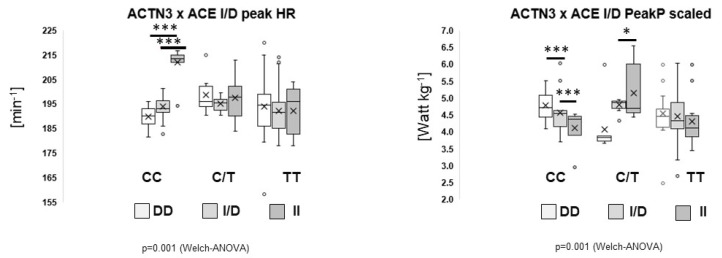
Main genotype interactions with cardiorespiratory parameters. Box-whisker plots of the association of peak heart rate and scaled peak power during CPX-testing with the interaction between the ACTN3 (rs1815739) and ACE-I/D (rs1799752) genotypes. ANOVA with post hoc test of least significant difference. Note: Post-hoc effects were identified between noncarriers and carriers of the I-allele (ACE) in CC genotypes of ACTN3 (rs1815739) for the maximal heart rate during the ramp test to exhaustion. Conversely, the scaled peak power output was lowest in the homogenous carriers of the I-allele (ACE) in CC genotypes. * and *** indicate, *p* ≤ 0.05 and 0.001, respectively, for the indicated comparisons.

**Table 1 genes-13-01798-t001:** Characteristics of the athletes. Summary of the genotypes of the studied alpine and nordic skiers from the Swiss skiing squad [27,28].

Gene	ACE-I/D			Tenascin C			ACTN3			PTK2			PTK2		
SNP	rs1799752			rs2104772			rs1815739			rs7843014			rs7460		
		n	percent		n	percent		n	percent		n	percent		n	percent
skiers	D/D	8	18.6	A/A	16	37.2	C/C	12	27.9	C/C	7	16.3	A/A	15	34.9
	D/I	24	55.8	A/T	23	53.5	C/T	6	14.0	C/T	20	46.5	A/T	23	53.5
	I/I	11	25.6	T/T	4	9.3	T/T	25	58.1	T/T	16	37.2	T/T	5	11.6
		43	100		43	100		43	100		43	100		43	100
*p*-value	0.425			0.294			<0.0001			0.859			0.391		
		n	percent		n	percent		n	percent		n	percent		n	percent
data points	D/D	65	28.3	A/A	77	33.5	C/C	59	25.7	C/C	38	16.5	A/A	85	37.0
	D/I	117	50.9	A/T	126	54.8	C/T	34	14.8	C/T	113	49.1	A/T	122	53.0
	I/I	48	20.9	T/T	27	11.7	T/T	137	59.6	T/T	79	34.3	T/T	23	10.0
		230	100		230	100		230	100		230	100		230	100
*p*-value	0.728			0.023			<0.0001			0.822			0.029		

**Table 2 genes-13-01798-t002:** Physiology of the studied athletes from the Swiss-ski squad. 230 data points were available, whereby a total of 168 and 62 data points were from alpine and Nordic athletes, respectively. 95 and 135 data points, respectively, were from male and female athletes.

Parameter	Unit	Mean		SD	[Min–Max]	N
age	[years]	21.5	±	3.02	[16.22–31.58]	230
body mas	[kg]	[kg]	±	10.9	[52.10–106.10]	230
body size	[cm]	173.99	±	8.69	[155.00–202.00]	230
peakVO_2_	[L(O_2_) min^−1^]	3.69	±	0.79	[2.35–5.62]	230
peakVO_2__scaled	[L(O_2_) min^−1^ kg^−1^]	0.05	±	0.01	[0.03–0.08]	230
peakRER	[L(CO_2_) × L(O_2_)^−1^]	1.52	±	0.18	[1.11–2.05]	230
peakP	[Watt]	320.99	±	61.74	[190.00–460.00]	230
peakP_scaled	[Watt × kg^−1^]	4.54	±	0.68	[2.48–6.54]	230
peakVE	[L min^−1^]	145.54	±	32.44	[66.09–227.45]	230
peakVT	[L]	2.65	±	0.53	[1.52–4.01]	230
peakHF	[beats min^−1^]	194.03	±	9.24	[158.20–220.00]	230
max_P_ext_R360	[Watt]	697.94	±	198.25	[412.40–1349.90]	141
max_P_ext_L360	[Watt]	673.81	±	169.75	[374.10–1167.80]	141
max_P_flex_R360	[Watt]	464.15	±	170.98	[217.60–1117.70]	141
max_P_flex_L360	[Watt]	450.5	±	148.9	[226.80–938.90]	141

## Data Availability

Data is available on serious request by the corresponding author.

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
