# Peer review of "Association of Gene Variants for Mechanical and Metabolic Muscle Quality with Cardiorespiratory and Muscular Variables Related to Performance in Skiing Athletes"

_genes, 2022, doi:10.3390/genes13101798_

Round 1
Reviewer 1 Report
Manuscript titled “Association of gene variants for mechanical and metabolic muscle quality with cardiorespiratory and muscular variables related to performance in skiing athletes” adds further knowledge on the association between DNA polymorphisms and sport performance. Particularly authors focus their attention on skiing, filling existing gaps in literature.
The manuscript is well developed, with adequate methods and statistics, so in my opinion should be accepted with minor revision.
In details:
1. Sampling should be more detailed, specifying how many males and females are included. 2. Sampling. Maybe I missed something, but it is not clear to me why in material and method authors speak about a sample of 43 and I have 239 genotypes in table 1. 3. Since sample is mixed (male+female), I suggest not to indicate the average for height and weight (both in text and table), since they are linked to gender. 4. Results: Authors report a deviation from HW equilibrium for ACTN3, but in table 1 also tenascin C and PTK2 rs7460 do not match the equilibrium. P value is significant and highly significant for ACTN3. If author fix the cut off for significant at 0.01, it should be specified. 5. Table 1. It would be better indicating the percentage and not the absolute number for the genotype frequencies; moreover, it would be useful to also indicate the frequency in “general” population. 6. Results and discussion. I am very perplexed about the frequency of ACTN3, since the prevalence of TT was never found in European population. I think it deserves further investigation, (for example an analysis of general population, or checking the sample to be sure that relatives are not included). Moreover, the strongly deviation from HW equilibrium should not be underestimated: it is due to skiing selection or to genetic drift? You can answer only testing the general population or using data from literature.
Minor revision:
There are many refused in the punctuation.
Author Response
a

Reviewer 2 Report
The article has a problem with the design of references to literature. In the annotation, it is necessary to indicate the pc numbers of the genes that were analyzed. Tests for athletes are very poorly described. The results do not indicate the statistical significance of the data obtained. The work hypothesis needs to be expanded. The originality of the work is 68.66%. I recommend the article for publication after correcting the comments.
Author Response
a

Round 2
Reviewer 2 Report
There are no comments. Thanks for the corrections!